Supernumerary teeth observed in a live True’s beaked whale in the Bay of Biscay

Robbins James R. james.robbins@orcaweb.org.uk jamesrichardrobbins@googlemail.com 1 2
Park Travis 3
Coombs Ellen J. 3 4
1 ORCA , Portsmouth , United Kingdom
2 School of Biological and Marine Sciences, University of Plymouth , Plymouth , United Kingdom
3 Department of Life sciences, Natural History Museum , London , United Kingdom
4 Department of Genetics, Evolution, and Environment, University College London, University of London , London , United Kingdom
Hopper Lydia
Electronic publication date: 2019 Oct 14
Publication date: 2019
Volume: 7
Electronic Location ID: e7809
Received 2019 May 28; Accepted 2019 Sep 1
Copyright: ©2019 Robbins et al.
Copyright year: 2019
Copyright holder: Robbins et al.
License: This is an open access article distributed under the terms of the Creative Commons Attribution License, which permits unrestricted use, distribution, reproduction and adaptation in any medium and for any purpose provided that it is properly attributed. For attribution, the original author(s), title, publication source (PeerJ) and either DOI or URL of the article must be cited.
License URL: https://creativecommons.org/licenses/by/4.0/

Keywords: True’s beaked whale, Mesoplodon mirus, Dentition, Bay of Biscay, Cetacea, Morpohology

Funding: Marie Skłodowska-Curie Individual Fellowship 748167/ECHO The London Natural Environment Research Council Doctoral Training Partnership NE/L002485/1 James Robbins was supported by ORCA, who are funded by memberships and supporters. This research was supported by a Marie Skłodowska-Curie Individual Fellowship (748167/ECHO) to Travis Park. The London Natural Environment Research Council Doctoral Training Partnership (London NERC DTP) training grant NE/L002485/1 supported Ellen Coombs. The funders had no role in study design, data collection and analysis, decision to publish, or preparation of the manuscript.

==============================
Mesoplodont beaked whales are one of the most enigmatic mammalian genera. We document a pod of four beaked whales in the Bay of Biscay breaching and tail slapping alongside a large passenger ferry. Photographs of the animals were independently reviewed by experts, and identified as True’s beaked whales (Mesoplodon mirus). This is the first conclusive live sighting of these animals in the north-east Atlantic, and adds information to previous sightings that are likely to have been M. mirus. Photographs of an adult male appears to show two supernumerary teeth posterior to the apical mandibular tusks. Whilst analysed museum specimens (n = 8) did not show evidence of alveoli in this location, there is evidence of vestigial teeth and variable dentition in many beaked whale species. This is the first such record of supernumerary teeth in True’s beaked whales.

Introduction

Beaked whales (Ziphiidae) are among the least understood cetacean families. Within this family, the genus Mesopolodon consists of approximately 14 species that are morphologically similar, and many remain relatively unknown (Pitman, 2009; Ellis & Mead, 2017). Species within this genus have a pelagic distribution and a preference for deep waters. Most of our knowledge of many of these species is from stranded specimens as they are rarely observed at sea due to long deep dives and short time spent at the surface. Four Mesoplodonts (Blainville’s, Mesoplodon densirostris; Gervais’, Mesoplodon europaeus; Sowerby’s, Mesoplodon bidens; and True’s beaked whales, Mesoplodon mirus) have been observed in the north Atlantic, however they are difficult to identify to species level from brief glimpses at sea (MacLeod, 2000; MacLeod et al., 2006).

Mesoplodonts are similar in size (3.9–6.2 m) and exhibit intraspecific and interspecific variation in colouration (Aguilar de Soto et al., 2017). The size and shape of the beak and melon aid identification (Jefferson, Webber & Pitman, 2015). In the North Atlantic, True’s beaked whales are most likely confused with Gervais’ beaked whales due to a shared grey colouration, dark eye patch, and pale ventral area. Both species have short, straight beaks (Ellis & Mead, 2017).

Most adult male ziphiids have a single pair of tusks, with the exception of Shepherd’s beaked whale (Tasmacetus) which has a full complement, and the genus Berardius which have two pairs of tusks. The position of tusks in the mandible is one of the most diagnostic characteristics for Mesoplodon identification (MacLeod, 2000; Weir et al., 2004). True’s beaked whales have two small apical tusks, relatively close together compared to other species. Gervais’ beaked whales also have a pair of teeth which are placed further posterior than those seen in True’s (Ellis & Mead, 2017). However, the teeth of females and subadult males do not erupt beyond the gum, so this identification characteristic is reliable only for adult males.

Due to the inherent difficulty of identifying Mesoplodon mirus at sea, there have been few sightings confirmed to species level. There have only been three published sightings of True’s beaked whales in the Bay of Biscay, most recently in 2003 (Weir et al., 2004). No erupted tusks were observed in these animals, and therefore identification was not definitive; however, a combination of identification features coincides with those of True’s beaked whales (McLellan et al., 2018).

The Bay of Biscay is a heterogeneous habitat with varied bathymetry that ranges from shallow coastal waters, continental shelf edge, deep pelagic, and deep subsea canyons and seamounts (Kiszka et al., 2007). This area supports a diverse range of cetacean species, including several beaked whale species (Matear et al., 2019). Cuvier’s beaked whales (Ziphius cavirostris) are frequently recorded (Kiszka et al., 2007; J Robbins, 2019, unpublished data), and Sowerby’s beaked whales and northern bottlenose whales (Hyperoodon ampullatus) are infrequently recorded (J Robbins, 2019, unpublished data).

In this study we describe a sighting of True’s beaked whales in the Bay of Biscay, showcase photographic evidence that likely represents the best quality images of live individuals in the north-east Atlantic, and analyse the photographs which indicate the presence of a second pair of teeth in one individual.

Methodology

In-situ data collection

Data were collected in the Bay of Biscay by ORCA, a citizen science charity (http://www.orcaweb.org.uk; Robbins, Babey & Embling, 2019) that have collected data in this area since 1995. Staff and trained volunteers were guiding a photography trip with Jessops Academy on the MV Pont-Aven ferry (184 m length, 24.1 m deck height) which transits between Portsmouth, UK–Santander, Spain–Plymouth, UK. Eleven people (three experienced with cetacean surveys, eight guests) were actively searching for marine megafauna. Location data were collected every second on a Microsoft Surface tablet with GPS dongle attached, and environmental conditions (sea state, swell, visibility, precipitation, glare) were collected at the beginning of the survey, and whenever conditions changed, or every 30 min at a minimum. All cetacean sightings were recorded, with accompanying photographs taken using an array of camera bodies and lenses.

Identification

Photographs of animals were circulated to seven independent experts for species identification.

Museum specimens

Museum specimens of True’s beaked whales were examined at the Natural History Museum (NHM; n = 1), and the Smithsonian National Museum of Natural History (USNM; n = 7). Specimens were of various ages and sexes and recovered from the east coast of the USA, and Ireland (Table 1). Signs of a second pair of tusks were investigated by looking for mandibular or vestigial teeth. The mandible was examined for alveoli which are indicative of mandibular tooth presence, whether erupted or not. Photographs were taken for reference, and 3D surface scans were taken using a Creaform Go!SCAN 50 laser scanner and VXElements software. Scans were cleaned, prepared, and exported to .ply in Geomagic Wrap software (3D Systems) at a resolution of 0.8 mm and then rendered in MeshLab.

Table 1 Information on the museum specimens of True’s beaked whales examined.

Holding institution	Specimen number	Sex	Age class	Collection date	Location	
Natural History Museum	NHMUK1920.5.20.1	Male	Adult	09/06/1917	County Clare, Ireland	
Smithsonian National Museum of Natural History	USNM504612	Female	Adult	13∕08∕1977	New Jersey, USA	
USNM504724	Male	Adult	03∕11∕1977	Maryland, USA	
USNM504764	Male	Juvenile	04∕03∕1978	New Jersey, USA	
USNM550351	Male	Unknown	02∕08∕1983	Washington, USA	
USNM571357	Male	Adult	26∕05∕1989	New Jersey, USA	
USNM571459	Female	Adult	12∕05∕1991	Delaware, USA	
USNM572961	Female	Adult	10∕10∕2003	North Carolina, USA	

Results

A group of beaked whales was observed on the 4th of July 2018, at 14:10:34 UTC in good conditions (Beaufort 3, light swell, good visibility). The group was estimated in-situ to consist of four individuals, and this was confirmed on reviewing 842 photographs taken during the encounter. It is possible that more animals were present but not observed at the surface. The species were independently identified by six of the seven contacted experts as True’s beaked whales, with the 7th being unsure but unable to offer a possible alternative species. Identification was based on the slope of the melon (Ellis & Mead, 2017), colouration (Figs. 1A–1C; Shirihai & Jarrett, 2007), the presence of parallel scars from rival males Fig. 1D; Aguilar de Soto et al., 2017; Ellis & Mead, 2017), and the two tusks placed towards the front of the mandible in one of the individuals (Fig. 1E; Ellis & Mead, 2017).

Figure 1 Morphological characteristics of True’s beaked whales, as photographed during the encounter in July 2018.

(A) Ventral surface lighter than dorsal and light anal patch. (B) Flipper pockets for streamlined diving. (C) Dark eye patch. (D) Parallel scars inflicted from close together tusks. (E) Two small tusks on the apical tip of the mandible, and a dark-tipped beak (photographed by Christopher Hogben).

The animals were spotted approximately 62 km north of Santander (44.05416 N, 3.92118 W; 3,000 m water depth; Fig. 2) in the southern Bay of Biscay, three km from the Torrelavega canyon (>4,100 m depth). Perhaps coincidentally, this is 13 km from where a group of three possible True’s beaked whales, including two adults and a calf, was recorded on an ORCA survey on the 11th of July 2007, at 07:40 UTC (44.172 N, 3.967117 W; 3,800 m water depth; Fig. 1); however, the sighting in 2007 was not corroborated by photographic evidence. All sightings in the Bay of Biscay have been in water over 2,000 m deep, with a mean water depth of 3,159 m.

Figure 2 Sightings of True’s beaked whales in the Bay of Biscay.

Previous sightings reported by Weir et al. (2004) (grey stars) are highly likely to be True’s beaked whales. The sighting in 2007 was not confirmed by experts or photographs but is displayed as the only other record in a database spanning 2006–2018 and is situated close to the confirmed sighting (red triangles). Bathymetry vector courtesy of Natural Earth: www.naturalearthdata.com.

Sighting description

With few confirmed sightings at sea, knowledge of the behaviour of this species is severely limited. We provide a brief description here.

The group was initially spotted parallel to the bow, and next surfaced approximately 15 m from the ship, midway along the port side, where three of them breached and landed on their side, and the fourth animal broke the surface with its beak in swimming behaviours often seen in beaked whales. The ship passed the animals and they breached three or four times in the wake of the ship, falling back to the water on their sides. Some animals cleared the water completely during these breaches. Breaching animals often curved their bodies whilst in the air (Fig. 3).

Figure 3 Sequence of photographs showing three True’s beaked whales breaching together during the encounter in July 2018.

Letters (A–F) represent the sequence photographs were taken. All photos were taken in less than two seconds. (Photo credit: Christopher Hogben).

The animals were not seen for several seconds and were seen again approximately 200 m from the ship. The group exhibited tail slapping behaviour (around 15–20 times) at the surface before breaching again, this time with a shallower exit angle more horizontal to the water than previously observed. These lower breaches sometimes ended with a mid-air twist by the animals so that they landed on their sides. In some instances, animals twisted so that they landed on their backs, with their ventral surface visibly lighter in colouration. There was no dive witnessed as the ship passed, with them last seen breaching behind the ship. The entire encounter lasted 68 s whilst the vessel travelled at 23.0 knots.

Morphological observations from live animals

Apical tusks are visible in several photos (Fig. 4), showing at least one individual is an adult male. There are two additional teeth visible posterior to the typical pair, which appear to be a similar distance from the anterior set as the apical pair from each other.

Figure 4 Dorsal view of a male True’s beaked whale photographed in July 2018, taken from two (A & B) marginally different angles.

Both images show an additional smaller pair of teeth posterior to the typical apical tusks. The crescent blowhole and a linear tooth rake are also visible. (Photo credit: Christopher Hogben).

In addition to the identifying characteristics previously mentioned, many individuals were strongly scarred, showing evidence of linear and parallel tooth rake marks. These are visible on the dorsal and ventral surfaces (Fig. 5). The anal-genital patch appeared light pink (Fig. 5B).

Figure 5 True’s beaked whale appearance, as photographed during the encounter in July 2018.

(A) Ventral view showing no medial fluke notch, an anal-genital patch with pink colouration, faint tooth rake scarring, and pectoral flipper pockets. (B) Dorsal view showing tusks, a crescent blowhole, and tooth rake scars. Scars include an extensive linear line leading from the blowhole backwards down the dorsal surface, several shorter linear lines, and two parallel lines at a similar spacing as the apical tusks. (A) and (B) may be of separate animals. (Photo credit: Christopher Hogben).

Museum specimens

There were no signs of large alveoli that could represent a second pair of mandibular teeth in any examined specimens. Unerupted teeth in females and subadult males were apparent, as were alveoli—suggesting that they would not be overlooked if present in museum specimens (Fig. 6). The vestigial alveolar groove was present on the mandible of all specimens.

Figure 6 Adult True’s beaked whale female mandible (USNM504612), showing alveoli around unerupted teeth at the apical tip of the jaw anterior to the vestigial alveolar groove.

Photographs are from anterior (A) and dorsal (B) viewpoints. Photo credits: Diane E. Pitassy and Don Hurlbert, National Museum of Natural History Imaging Services, Smithsonian Institution.

It was not possible to measure the aperture of alveoli due to the poor condition of mandibles. Only one specimen (NHMUK1920.5.20.1) had a complete skull in good condition, although tusks shown are casts of the originals. We reproduce photographs of this specimen below (Figs. 7 and 8) to add to the limited published records, and 3D scans have been archived on the Phenome10K website (http://phenome10k.org/; Goswami, 2015).

Discussion

Mesoplodonts are notoriously difficult to identify, especially at sea. True’s beaked whales are most commonly confused with Gervais’ beaked whales, which can be ruled out as images were independently identified on a range of characteristics, most prominently being the position of tusks at the tip of the mandible as opposed to 1/3 of the way from the tip to the corner of the mouth (Ellis & Mead, 2017). These photos therefore represent the first confirmed sighting of True’s beaked whales alive in the north-east Atlantic.

True’s beaked whales are rarely observed at sea, and rarely strand, with only 12 records in the UK and Ireland between 1917 and 2013, all of which were in Ireland (Natural History Museum, 2018; Coombs et al., 2019). There have been several live sightings in the Bay of Biscay that are likely to be True’s beaked whales; although photographs are not conclusive (Weir et al., 2004). Information on the behaviour of True’s beaked whales is scarce due to confirmed sightings of live animals being limited. The accounts published here provide a small but relatively important insight into the behaviour of True’s beaked whales. Animals were observed breaching in quick succession in a close formation, similar to that filmed in Gervais’ beaked whales (Aguilar de Soto et al., 2017).

Figure 7 Dorsal view of an adult male True’s beaked whale skull (NHMUK1920.5.20.1), showing two large apical tusks.

Tusks are casts of the originals. Photo credit: Travis Park.

Figure 8 Lateral view of apical tusks in a male True’s beaked whale (NHMUK1920.5.20.1).

The background tusk is complete, whereas the foreground one is worn and/or damaged. Both tusks are casts of originals. The vestigial alveolar groove is present posterior to the tusks. (Photo credit: Travis Park).

Each sighting of True’s beaked whales can provide key information on areas of importance, distribution, and variation between individuals. This confirmed sighting and previous sightings likely to be the same species highlight the Bay of Biscay as a suitable habitat for this species that is worth conserving. There are several beaked whale species frequently observed in the area which utilise similar habitats, and this record provides evidence for another. To date, all potential sightings of True’s beaked whales in the Bay of Biscay have been from platforms of opportunity, namely ferries. Beaked whales are often inconspicuous and elevated observation platforms available on ferries are unlikely to often provide views good enough for identification of Mesoplodon species. Based on the findings of this study, and those in the literature (e.g., Weir et al., 2004; Kiszka et al., 2007; Dolman et al., 2011; Laran et al., 2017), a dedicated survey researching beaked whales in the Bay of Biscay could further investigate species diversity and occurrence.

The photographed evidence of an additional pair of teeth is unexpected, with no previous records of this dentition in other live or stranded animals. There was also no evidence of additional impacted or unerupted teeth in the museum specimens analysed; however small vestigial teeth have been recorded in several other ziphiids (Boschma, 1951; Kirino, 1956; Fordyce, Mattlin & Wilson, 1979; Gomercie et al., 2006). Vestigial teeth may not appear in skeletal specimens as they may not be set in alveoli, but instead attach loosely in the gum (Gomercie et al., 2006). Whilst the photographs are not clear enough to rule out the posterior set being vestigial, they appear only slightly smaller than the apical tusks, suggesting that they may be mandibular. However, a stranded Cuvier’s beaked whale was recorded with a similarly placed, single supernumerary mandibular tooth that was also smaller than the apical tusks, but was not set in alveoli (Fordyce, Mattlin & Wilson, 1979).

It is likely this male True’s beaked whale represents the first record of supernumerary teeth in True’s beaked whales. Our sample size of examined museum specimens is relatively small as a result of the rarity of the species in museum collections. As this condition has not been reported previously, and is rarely reported in more common ziphiids, it is logical to assume that only a small percentage of animals possess supernumerary teeth. Alveoli may be missed if not looked for, and therefore we suggest that holders of True’s beaked whale mandibles investigate this further. Individuals that strand in the future should be examined closely for this condition. There is still a dearth of information on Mesoplodonts and additional records are important to our understanding of this rarely seen family.

This may also be the first photograph of a pink anal patch in a live True’s beaked whale. All available literature suggests patches are white (Weir et al., 2004); although it is common for colouration to appear different after death (Stockin et al., 2009; Aguilar de Soto et al., 2017), and it has been hypothesized that pink ventral colouration may exist in live Mesoplodon mirus (Raven, 1937). Other cetacean species have been recorded with variation in ventral colouration, such as Tucuxi (Sotalia fluviatilis), with pink colourations likely due to increased blood supply during increased activity (Edwards & Schnell, 2001).

Conclusion

This is the first conclusive True’s beaked whale sighting in the north-east Atlantic and builds upon previous Mesoplodon sightings that are likely to have been True’s beaked whales in the Bay of Biscay. This record presents new details on this data deficient species, their appearance and behaviour. The accompanying photographs are likely to be some of the best ever taken of live animals and show a previously unrecorded pink colouration in the genital-anal patch, and an additional pair of teeth not previously documented in stranded animals or museum specimens.

Supplemental Information

Supplemental Information 1 Dorsal view of a True’s beaked whale skull (NHMUK1920.5.20.1)

Captured in 3D using a Creaform Go!SCAN 50 laser scanner and VXElements software. Scans were cleaned, prepared, and exported to .ply in Geomagic Wrap software at a resolution of 0.8 mm and then rendered in MeshLab. The .ply file is available as supplementary material 2.

Click here for additional data file.

Supplemental Information 2 3D scan of Mesolodon mirus skull

Specimen NHMUK1920.5.20.1 captured in 3D using a Creaform Go!SCAN 50 laser scanner and VXElements software. Scans were cleaned, prepared, and exported to .ply in Geomagic Wrap software at a resolution of 0.8 mm and then rendered in MeshLab.

Additional files have been archived on the Phenome10K website (http://phenome10k.org; Goswami, 2015).

Click here for additional data file.

Thanks are due to Brittany Ferries for welcoming the group onboard, and to Jessops Academy for facilitating the trip. Steve Jones and Kate Weston from ORCA provided detailed accounts of the sighting. Brian Clasper (ORCA surveyor) and Jessops Academy attendees Christopher, Denise, Elaine, Ellie, Karen, Kate, and Robert provided photographs for use in this study. Lucy Babey proofread an early manuscript. Thank you to the Natural History Museum and Smithsonian for allowing access to their museum specimens, curators Richard Sabin (Natural History Museum), and Morgan Churchill (Department of Biology, University of Wisconsin Oshkosh) for allowing access to 3D scan data, and to the countless researchers who have prepared these specimens for use. Bay of Biscay regulars Paul Burley, Elfyn Pugh, Stephen Marsh, and John Young and cetacean and beaked whale experts Mark Carwardine, Robert Pitman, and James Mead provided expert opinions and independent reviews of photos for species identification.

Additional Information and Declarations

Competing Interests

Author Contributions

Animal Ethics

Data Availability

The authors declare there are no competing interests.

James R. Robbins conceived and designed the experiments, performed the experiments, prepared figures and/or tables, authored or reviewed drafts of the paper, approved the final draft.

Travis Park conceived and designed the experiments, performed the experiments, authored or reviewed drafts of the paper, approved the final draft, collected photographs and measurements from museum specimens.

Ellen J. Coombs conceived and designed the experiments, performed the experiments, prepared figures and/or tables, authored or reviewed drafts of the paper, approved the final draft, collected photographs, 3D scans, and measurements from museum specimens.

The following information was supplied relating to ethical approvals (i.e., approving body and any reference numbers):

All specimens were observed during opportunistic encounters from a ferry, or in existing museum collections so no additional approval was required.

The following information was supplied regarding data availability:

3D scans of True’s beaked whale skulls are available on Phenome10K under “Mesoplodon mirus”: http://phenome10k.org/mesoplodon-mirus/ (NHM specimen with tusks: Mesoplodon mirus NHM 1920.5.20.1)

http://phenome10k.org/mesoplodon-mirus-2/ (USNM specimen: USNM 571357)

http://phenome10k.org/mesoplodon-mirus-3/ (USNM specimen: USNM 504612)

The 3D scan of the True’s beaked whale skull is also available as a Supplemental File.

Specimen NHMUK1920.5.20.1 is available at the Natural History Museum (Ireland) and specimens USNM504612, USNM504724, USNM504764, USNM550351, USNM571357, USNM571459, and USNM572961 are available at the Smithsonian National Museum of Natural History (USA). All specimen locations are outlined in detail in Table 1.

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
