# Peer review of "Supernumerary teeth observed in a live True’s beaked whale in the Bay of Biscay"

_PeerJ, doi:10.7717/peerj.7809_

## Round 0.1 · original submission · Minor Revisions

I am sorry that it took me a while to obtain reviews for your article, but I have now received two reviews from experts in your field and provide their comments below. Both reviewers have kindly provided detailed and thoughtful comments on your article. As you revise your article, please be sure to respond to each of their comments providing details of what edits you have made to your article.

Both reviewers note the merit of your report but differ in their appraisal of which element of your sighting (the species, the teeth) is of key interest to readers. I will allow you to determine whether or not to re-frame your article accordingly, but please provide commentary to the reviewers as to your reasoning either way.

A couple of small points:

1) I note that reviewer 2 requests that you cut some of the information you provide regarding data collection (line 50) however I think that detail is useful and encourage you to leave it in.

2) On line 69 you use the term "genders", please replace this with "sexes".

·

Basic reporting

Be consistent throughout the ms between apical teeth and tusks. The apical teeth are now commonly referred to as “tusks” while additional teeth are called “teeth”

Be consistent in using mandible(s) instead of “jaw”. Jaws can be upper and/or lower, while mandible is anatomically correct.

Line by line edits:
Line 28: melon and beak “aid in identification” but it is teeth placement that confirms species ID as stated in the next paragraph.

Line 44: add McLellan et at 2018 (citation included here) that provided photographic identification of M. mirus, M. europeaus and Z. cavirostris and describes species ID characters delineating each. This publication also describes overlapping distributions and abundances of these three species.

McLellan, W.A., McAlarney, R.J., Cumming, E.W., Read, A.J., Paxton, C.G.M., Bell J.T., and Pabst, D.A. 2018. Distribution and abundance of beaked whales (Family Ziphiidae) off Cape Hatteras, North Carolina, USA. Marine Mammal Science. DOI: 10.1111/mms.1250 34(4): 997-1017

Line 52: add Pont-Aven “ferry”
Line 61: to independent experts “for species identification”
Line 93: how far off was the initial sighting? An initial ID of Tursiops is probably not important here and could be dropped as beaked whale sightings were not confirmed until photographs were analyzed?
Line 120: you only personally examined osteological specimens in EU collections? There are a number of complete skulls in collections in the US, so would question this finding.
Lines 130-135: There is no possibility of misidentifying the photos included in this ms as Berardiu safter expert analysis. This section could be dropped.
Lines 148-150: Nowhere in this ms are other beaked whale species described as having been sighted in the Bay of Biscayne.
Lines 168-170: did that specimen have an alveoli associated with that tooth placement?

Experimental design

No additional comments on the experimental design.

Validity of the findings

Lines 40-44. The confirmed sightings of M. mirus is the more important take home message of this ms. If the species was not confirmed until these observations (though I did not conduct an exhaustive literature search for beaked whales sightings in the Bay of Biscayne) that is more important than supernumerary teeth. The supernumerary teeth can be an additional observation in this ms, but a redirection of the ms would also require a change to the title “Confirming True’s Beaked Whale Distribution in the Bay of Biscayne” or something to that effect.

You appear confident that the sighting contained 4 animals. Beaked whales take long extended deep dives and shorter “bounce” dives so they display a large range of distances traveled under the surface. Following “normal” rules for lumping dolphins together as a single sighting are difficult to transfer to diving beaked whales. There is growing evidence that beaked whales are diving and rejoining other animals at the surface in a constant conveyor of animals from the bottom to the surface. It would be worth adding a stamen about the confidence you have to combine these separate surfacings into a sighting of 4 individuals.

The ms is based on two(?) images of a male True’s beaked whale with possible supernumerary teeth. As there are a number of specimens in museum collections that you reference in Table 1, it would be a valuable addition to go into the collection records and images of those specimens to determine if very small teeth were present at necropsy/dissection. As these teeth may be embedded only in the hypodermis/epidermis there would be no record of them in the osteological specimen.

Additional comments

There is little value added between Fig 1 and Fig 5. As Fig 5 shows most of the same landmarks Fig 1 could be dropped.

Fig 3 needs to have the multiple images better lined up, numbers should be changed to letters and have a color change to be better delineated.

The pink coloration of the genital patch is almost certainly evidence of heat dumping.

Reviewer 2 ·

Basic reporting

When framing their observations/putting them in context the authors have missed important references regarding Mesoplodon sightings confirmed to species. How this can be addressed is noted in the comments to the authors.

Experimental design

No comment

Validity of the findings

The paragraph discussing the importance of platforms of opportunity is not warranted given this manuscript presents a single 68-second sighting of a species known (from strandings) to be in the northeast Atlantic.

Additional comments

As the authors note, beaked whales are a poorly known group and Mesoplodon in particular are a poorly-known yet diverse genus. True's beaked whales have been conclusively recorded as stranded specimens in the Northeast Atlantic, and there are a number of sightings of live individuals that are likely this species, but this manuscript provides photo-documentation of live individuals. The importance of that in itself is perhaps somewhat overstated (given they are known in the general area from strandings), but this manuscript also documents supernumerary teeth in a live True's beaked whale, which is worth reporting. That said, the manuscript is overly long given the content and could be shortened by 25% without losing anything important. Some of the context of the rarity of sightings of confirmed species of Mesoplodon also needs addressing (see below).

22-23 and 40-41. I understand that the authors want to emphasize how unusual their sighting is but the importance/frequency of confirmed species sightings of Mesoplodon is overstated and needs to be toned down. The statement that "most of our knowledge is from stranded specimens" is certainly true for most species of Mesoplodon, but not at all true for M. densirostris, with long-term in-depth studies of that species in both the Pacific and multiple locations in the Atlantic. The statement "there have been few sightings confirmed to species level" when referring to Mesoplodon is clearly not correct - there have been hundreds of confirmed sightings of M. densirostris (see papers from the Bahamas, Canary Islands and Hawaii), as well as numerous sightings of M. bidens, M. layardii, M. europaeus, M. peruvianus etc.

Methods

50-52. There are a lot of unnecessary details- should delete "a citizen science charity... staff and trained volunteers guiding a photography trip with Jessops Academy".

61-65. This is all in the acknowledgements and does not need to be repeated here.

76/85/93 etc. The term "pod" should be replaced with "group" throughout. Pod implies genetic relatedness (think peas in a pod), and should not be used when group would suffice.

76. "was recorded" should be "was observed"

96/97 - the ship passed the animals - saying "the animals passed the ship" implies the ship was motionless and the animals swam by.

113/114. Just note "The anal-genital patch appeared a light pink" - the rest of the information in that sentence belongs in the Discussion.

137/139. This implies that there have been no strandings in the western North Atlantic, yet Table 1 shows 7, and the NOAA stock assessment report notes four strandings in the US over a more recent 5-year period https://www.fisheries.noaa.gov/national/marine-mammal-protection/marine-mammal-stock-assessment-reports-species-stock#cetaceans---small-whales - as well as bycatch records. The authors should also see DeAngelis et al. 2018 "A description of echolocation clicks recorded in the presence of True's beaked whale" JASA, and McLellan et al. 2018 "Distribution and abundance of beaked whales off Cape Hatteras, North Carolina, USA" MMS

146-160. While it is clear that platforms of opportunity provide valuable information, given this manuscript is about a single sighting having a long paragraph extolling the value of such platforms is a bit excessive. This paragraph could be reduced to one or two sentences.

---

## Round 0.2 · accepted · Accept

Thank you for your thorough responses to the reviewers' comments and suggested edits. It is my pleasure to accept your article for publication in PeerJ.